# A New Spectral Index for Monitoring Leaf Area Index of Winter Oilseed Rape (*Brassica napus* L.) under Different Coverage Methods and Nitrogen Treatments

**DOI:** 10.3390/plants13141901

**Published:** 2024-07-10

**Authors:** Hao Liu, Youzhen Xiang, Junying Chen, Yuxiao Wu, Ruiqi Du, Zijun Tang, Ning Yang, Hongzhao Shi, Zhijun Li, Fucang Zhang

**Affiliations:** 1Key Laboratory of Agricultural Soil and Water Engineering in Arid and Semiarid Areas, Ministry of Education, Northwest A&F University, Yangling 712100, China; hliunwafu@163.com (H.L.); junyingchen@nwafu.edu.cn (J.C.); phyllis12@nwafu.edu.cn (Y.W.); duruiqi@126.com (R.D.); tangzijun@nwsuaf.edu.cn (Z.T.); yangning1228@126.com (N.Y.); shihongzhao7@nwsuaf.edu.cn (H.S.); lizhij@nwsuaf.edu.cn (Z.L.); zhangfc@nwsuaf.edu.cn (F.Z.); 2College of Water Resources and Architectural Engineering, Northwest A&F University, Yangling 712100, China

**Keywords:** leaf area index, multi-angle hyperspectral, machine learning, winter oilseed rape (*Brassica napus* L.)

## Abstract

The leaf area index (LAI) is a crucial physiological indicator of crop growth. This paper introduces a new spectral index to overcome angle effects in estimating the LAI of crops. This study quantitatively analyzes the relationship between LAI and multi-angle hyperspectral reflectance from the canopy of winter oilseed rape (*Brassica napus* L.) at various growth stages, nitrogen application levels and coverage methods. The angular stability of 16 traditional vegetation indices (VIs) for monitoring the LAI was tested under nine view zenith angles (VZAs). These multi-angle VIs were input into machine learning models including support vector machine (SVM), eXtreme gradient boosting (XGBoost), and Random Forest (RF) to determine the optimal monitoring strategy. The results indicated that the back-scattering direction outperformed the vertical and forward-scattering direction in terms of monitoring the LAI. In the solar principal plane (SPP), EVI-1 and REP showed angle stability and high accuracy in monitoring the LAI. Nevertheless, this relationship was influenced by experimental conditions and growth stages. Compared with traditional VIs, the observation perspective insensitivity vegetation index (OPIVI) had the highest correlation with the LAI (r = 0.77–0.85). The linear regression model based on single-angle OPIVI was most accurate at −15° (R^2^ = 0.71). The LAI monitoring achieved using a multi-angle OPIVI-RF model had the higher accuracy, with an R^2^ of 0.77 and with a root mean square error (RMSE) of 0.38 cm^2^·cm^−2^. This study provides valuable insights for selecting VIs that overcome the angle effect in future drone and satellite applications.

## 1. Introduction

The leaf area index (LAI) is a crucial parameter for characterizing plant canopy structure, influencing processes such as light interception, respiration, transpiration, and net primary productivity [1]. The LAI reflects the dynamic changes in growth characteristics, canopy light distribution, respiration, photosynthesis, water vapor release, and carbon cycling [2]. Traditional methods for monitoring crop LAI, such as length–width factor method, lattice method, paper weight method, and laser leaf area method, can provide accurate measurements for small areas. However, these methods are time consuming, destructive, and unsuitable for large-scale monitoring [3]. Hyperspectral remote sensing emerged as a prominent tool in precision agriculture for monitoring crop growth, offering robust spectral continuity and high spectral information content for real-time and extensive LAI monitoring [4].

The vegetation canopy exhibits significant vertical gradients in biochemical and biophysical properties, posing challenges for traditional remote sensing techniques that rely on vertical observations [5]. Multi-angle remote sensing addresses these challenges by capturing comprehensive geometrical and spatial distribution information from multiple directions. This approach is widely used to derive crop growth and nutritional status from remote sensing data [6,7]. Hasegawa et al. [8] improved the accuracy of LAI inversion and mitigated saturation effects associated with the normalized difference vegetation index (NDVI) by integrating traditional vertical angle vegetation indices with hotspot and dark point information from multi-angle remote sensing. Similarly, Stagakis et al. [9] compared the ability of different view zenith angles (VZAs) to invert the LAI using satellite imagery data and ground-measured LAI data from 64 vegetation indices (VIs). These studies collectively highlight that non-vertical observations offer more reliable insights into canopy structural information compared to vertical observations, thus reducing errors in LAI inversion associated with canopy structural features [10].

High reflectance in the near-infrared (NIR) band indicates multiple scattering within plant leaf blades and canopy leaves, making it a reliable indicator of the LAI [11]. Extensive research has compared the stability and accuracy of various VIs for LAI estimation, optimizing them to enhance their linear correlations [12,13,14]. For instance, the widely used NDVI has limitations such as saturation and non-linear relationships in dense canopies and vigorous growth conditions [15]. To account for soil background effects, soil-corrected VIs like the soil-adjusted vegetation index (SAVI), modified soil-adjusted vegetation index (MSAVI), and optimized soil-adjusted vegetation index (OSAVI) have been developed [16,17,18]. The meris terrestrial chlorophyll index (MTCI), involving green and red edges, can alleviate LAI saturation [19]. Introducing the blue band into the NDVI helps mitigate the impact of atmospheric and surface factors [20]. These established VIs maximize sensitivity to LAI changes while minimizing adverse effects from soil, light, and atmosphere. However, as VIs are derived from the reflectance of multiple bands through various combinations, they exhibit varying sensitivities to view zenith angles (VZAs). For example, the NDVI generally shows higher values at larger VZAs, whereas the enhanced vegetation index (EVI) displays stronger angle sensitivity [10,21]. Galvão et al. [22] proposed that the EVI and photochemical reflectance index (PRI) exhibit strong anisotropy, whereas the NDVI and the Vogelmann index (VOG) show weaker angle sensitivities. Verrelst et al. [23] suggested that VIs constructed from bands between 450 and 1050 nm display significantly different angle sensitivities depending on crop types. Hovi et al. [24] emphasized that canopy structure significantly influences reflectance variations at different VZAs. Most of these studies have focused on analyzing the angle sensitivity of VIs observed vertically, with limited research utilizing multi-angle remote sensing data for crop growth monitoring. Therefore, it is necessary to analyze the anisotropic characteristics of crop canopies to reduce the impact of VZAs on monitoring crop growth and to develop novel multi-angle VIs. He et al. [25] improved the prediction accuracy and stability of leaf nitrogen content (LNC) by introducing a new angle-insensitive vegetation index (AIVI). Li et al. [26] developed an index algorithm based on multi-angle reflectance factors for leaf surfaces, which is effective for measuring leaf water content and applicable to various plant species with different significant and moisture contents under different measurement conditions. Currently, there is a lack of systematic analysis of the relationship between multi-angle VIs and the LAI, particularly for oilseed crops like rapeseed. This gap severely limits the application of various spectral analysis methods in multi-angle remote sensing. Future studies should focus on analyzing the angle sensitivity of different spectral analysis methods to identify VIs and techniques most sensitive to the LAI while being least sensitive to VZAs.

Oilseed rape is a widely cultivated essential oilseed crop globally, including in China, and its growth and yield formation are significantly influenced by nitrogen nutrition. The real-time, accurate, and rapid estimation of rapeseed LAI is essential for diagnosing and managing its growth and predicting yields [27]. This study was conducted to design field experiments involving various coverage methods and levels of nitrogen fertilization, with the aim of analyzing parameter changes under different VZAs. The goal is to develop VIs that exhibit stable performance under different VZAs and experimental conditions. Additionally, the optimal VI is inputted into various machine learning models to compare the monitoring accuracy of each model. This research provides technical guidance and a theoretical foundation for enhancing ground-based technologies in multi-angle remote sensing applications.

## 2. Results

### 2.1. Changes in the Canopy Reflectance and VIs of Feature Bands at Different VZAs

In this study, we selected five representative feature spectral bands: blue (450 nm), green (550 nm), red (660 nm), red edge (720 nm), and NIR bands (780 nm). As illustrated in Figure 1, visible light exhibits a significantly asymmetrical shape within the SPP, while the red edge and NIR bands are almost symmetrical. For visible light, the reflectance of blue and red bands shows minimal change with the varying VZA, with amplitudes within 40%. In contrast, the green and NIR bands exhibit higher amplitudes, reaching up to 122%. Generally, in the back-scatter direction, the reflectance of all five feature bands increases with an increase in VZA, with the minimum reflectance occurring near the vertical angle.

The changes in 16 VIs constructed from these five feature bands with respect to VZAs show fluctuations. Table 1 presents the change amplitude and F significance analysis of VIs relative to the vertical angle. A larger F-value indicates greater variation within that angular range. The results show that all the parameter variations exhibit significant differences at the *p* < 0.05 level in the range of −60° to +60°, while all the other parameters except the PRI, DVI, NDDA, VOG-2, DD, and CCII did not show significant differences at the *p* < 0.05 level in the range of +30° to +60° VZA. As shown in Figure 2, all parameters exhibit significant angular changes, with the NDDA showing the most prominent change between the forward- and back-scatter direction, with amplitudes reaching −29.77% and 95.42%. The F-test revealed significant difference at the *p* < 0.01 level in the range of −60° to +60° and +30° to +60°, indicating that the NDDA is the most sensitive parameter to VZAs. In the forward-scatter direction, most parameters show gradual stabilization after +30°, while the NDRE, NDDA, VOG-2, and CCII continue to increase with increasing VZAs. In the back-scatter direction, most parameters show a sharp increase or decrease after −15°, with the SAVI, NDDA, and DD being the most obvious. Additionally, the PRI, NDDA, and DD undergo sudden changes at a VZA of −45°. R1-dB, mND705, DDn, and DD exhibit a symmetrical shape in both the forward- and back-scatter direction, with their minimum values occurring at the vertical angle and increasing with VZAs. In summary, Vlopt, the OPIVI, and REP exhibit no significant change with the respect to VZAs and show a gentle angular change trend within the SPP.

### 2.2. Relationship between LAI and VIs under Different VZAs

This study comprehensively analyzes the relationship between VIs and the LAI based on the performance of the same spectral parameter under different VZAs and the performance of different VIs under the same VZAs. A linear regression model was established to monitor the LAI using VIs under different VZAs. As shown in Figure 3, VIs have a closer relationship with the LAI in the back-scatter direction. Among the 16 parameters, MTCI exhibited the poorest regression potential (R^2^ = 0.09–0.45) for all VZAs. Additionally, VIs such as the NDRE, NDDA, VOG-2, and CCII also showed poorer regression potential. For two-band VIs, although the PRI, SAVI, and DVI maintain a good correlation with the LAI near the vertical angle, their correlation coefficients decrease with increasing VZAs in both the forward- and back-scatter direction, indicating poor stability. In contrast, among the three-band VIs, mND705, EVI-1, and OPIVI exhibit strong correlations with the LAI. However, considering angular stability, the OPIVI performs the better. Among the four-band VIs, the REP shows high potential for monitoring crop LAI. Overall, most parameters are sensitive to changes in VZAs. As shown in Figure 4, EVI-1, the OPIVI, and the REP maintain high monitoring accuracy while demonstrating good angular stability. Notably, the OPIVI performed the best in the single-angle linear regression model at a VZA of −15°, with an R ^2^ of 0.71 and RMSE of 0.55 cm^2^·cm^−2^.

### 2.3. Comparison of VIs in Different Experimental Conditions across All VZAs

Numerous studies have confirmed that different experimental conditions can affect the angular sensitivity of VIs, necessitating an analysis of how different reproductive stages and treatments impact VIs’ angle sensitivity. As shown in Table 2, the correlation between different VIs and the LAI varies with different experimental factors. The R values under different experimental conditions show similar trends, with higher R values for vertical angles than for forward- and back-scatter direction. As shown in Figure 5, the parameters exhibit varying levels of correlation and angular stability under different experimental factors. During various growth stages, both the OPIVI and REP parameters demonstrate good angular stability while maintaining a high correlation with the LAI. However, there are significant differences in the correlation between EVI-1 and the LAI during the budding and flowering stages. Different treatments also results in significant differences in the correlation between the EVI-1 and REP parameters when comparing coverage method treatment and nitrogen fertilizer treatment.

Given the significant differences among experimental conditions (Table 2), the difference of R (DR, the difference in R among different experimental conditions) were calculated.
(1)$$DR=\frac{R_{\max}-R_{\min}}{R_{\mathrm{average}}}$$

Equation (1) demonstrates that the value of *DR* can reflect the angular stability of the correlation between VIs and VZAs. A smaller *DR* value indicates higher angular stability of the relationship between VIs and the LAI. The *DR* values were compared across all experimental factors. The results showed that the OPIVI had the strongest correlation with the LAI and exhibited the most stable change under different VZAs, followed by the REP and EVI-1. Specifically, the correlation coefficient for the OPIVI was slightly higher during flowering stage (0.85) compared to the seedling stages (0.82). Additionally, the correlation coefficient for cover treatment (0.86) was marginally higher than for nitrogen fertilizer treatment (0.82).

### 2.4. Estimating LAI by Different Machine Learning Algorithm

The three best angles (−30°, −15°, and 0°) of the OPIVI were selected as independent variables and input into three machine learning algorithms, SVM, XGBoost, and RF, for modeling. As shown in Figure 6, the results varied in monitoring accuracy among the algorithms. The RF algorithm performed the best, with an R^2^ of 0.77 and an RMSE of 0.38 cm^2^·cm^−2^, indicating that the Bagging model within the ensemble algorithm effectively utilized the information from the multi-angle VI data. The XGBoost algorithm achieved an R^2^ of 0.73 and an RMSE of 0.41 cm^2^·cm^−2^, slightly inferior to the RF algorithm. The SVM had the poorest performance in the multi-angle OPIVI monitoring of winter oilseed rape LAI (R^2^ = 0.71, RMSE = 0.51 cm^2^·cm^−2^). Overall, the RF algorithm, as an ensemble algorithm with integrated thinking, demonstrates strong comprehensive application capabilities and a high utilization rate of effective information in multi-angle VI data. It can serve as a robust algorithm support for monitoring crop nutrient parameters using multi-angle remote sensing data.

## 3. Materials and Methods

### 3.1. Experimental Design

This experiment was conducted at the Key Laboratory of Agricultural Soil and Water Engineering in Arid and Semiarid Areas of the Ministry of Education, Northwest A&F University, Yangling, from October 2022 to June 2023. The study area is a typical semi-humid and drought-prone region with a warm temperate semi-humid monsoon climate. The winter rapeseed variety used in this experiment is “Shaanyou No.18”. A total of 45 plots were used for data collection in this experiment, each with a size of 4 m × 6 m (24 m^2^) and arranged randomly. The experiment included five nitrogen (N) application rates: N0 (0), N1 (70 kg/hm^2^), N2 (140 kg/hm^2^), N3 (210 kg/hm^2^), and N4 (280 kg/hm^2^). Additionally, three types of mulching treatments were applied: straw mulching (SM) for flat crops, film mulching (FM) for ridges, and no mulching (NM) for flat crops. This resulted in a total of 15 treatments (Table 3), replicated three times. A 2 m wide protective belt surrounded the experimental area.

### 3.2. Measuring Multi-Angular Spectra and LAI

The spectral reflectance of winter oilseed rape (*Brassica napus* L.) canopy was measured using an ASD field-spec 4 back-mounted field spectrometer (LICA United Technology Limited in Beijing, China) with the field gonimeter system (FIGOS) as the reference [28]. A multi-angle hyperspectral monitoring device was designed to meet specific requirements (Figure 7a,b), ensuring the same target was observed at different VZAs. Measurements were taken between 11:00 and 13:00 under clear, windless conditions with good visibility.

For multi-angle hyperspectral data collection, the optical fiber is first fixed on the rocker arm of the bracket positioned at a height of 1.5 m. The real-time sun azimuth was obtained using an open-source sun azimuth calculator. The horizontal azimuth of the bracket was adjusted to ensure that the measurement plane was within the solar principal plane (SPP). Different VZAs were achieved by controlling the rocker arm, which moved back and forth to ensure consistent observations of the same area within the plot. The observation object was marked for accuracy. VZA was defined as 0° at vertical monitoring. The direction of sunlight opposite to the observation direction was defined as the forward-scatter direction (+), while the same side as the observation direction was defined as the back-scatter direction (−). VZAs were arranged as −60°, −45°, −30°, −15°, 0°, 15°, 30°, 45°, and 60° (Figure 7). Three measurements were taken at each VZA, and their average value was used as the spectral reflectance at that angle. Reference board calibration was performed immediately before and after each of VZA measurement (with the reflectance board having a reflectance of 1).

Following spectral data collection, three representative winter oilseed rape plants were randomly selected from each plot. These plants were separated into stems and leaves, and the leaf area was determined using the threshold segmentation of the photographs. The LAI was then calculated by multiplying the mean leaf area of the three plants by the number of single stems per unit area (obtained from field surveys conducted during critical growth stages).

### 3.3. Construction of the New Vegetation Index

Numerous studies have indicated that the NDVI is a frequently used and well-inverted spectral parameter. The saturation phenomenon of the parameters can be alleviated by introducing new bands or fixed coefficients based on the NDVI [29]. Considering that the blue and red bands are insensitive to changes in VZAs, and they are closely related to the leaf area index, it is proposed to introduce these bands into the NDVI. The specific formula is as follows:(2)$$\mathrm{OPIVI}=\frac{R_{\lambda 1}-R_{\lambda 3}}{R_{\lambda 2}-R_{\lambda 3}}$$

Among them, *λ*1 refers to all red edge bands within the range of 700–760 nm. *λ*2 and *λ*3 represent red and blue bands, respectively. After screening and comparison, it is determined that *λ*1 = 720 nm, *λ*2 = 660 nm, and *λ*3 = 450 nm. The specific forms are as follows:(3)$$\mathrm{OPIVI}=\frac{R_{720}-R_{450}}{R_{660}-R_{450}}$$

### 3.4. Data Analysis

#### 3.4.1. Preprocessing of Spectral Data and Construction of VIs

Numerous studies have indicated that the wavelength range of 350–1300 nm is particularly sensitive and characteristic spectral for reflecting crop pigments, nutrients, and the overall growth and development status [30]. Consequently, this study utilizes this wavelength region to analyze the LAI of winter oilseed rape. To reduce the influence of background noise, baseline drift, and undesirable elements such as scattered light on hyperspectral reflectance, preprocessing techniques are employed. These techniques include Savitzky–Golay convolution smoothing and quadratic polynomial function fitting and filtering for denoising [31].

The VI inversion method is a well-established approach for parameter inversion. Table 4 lists the commonly used two-band, three-band, and four-band VIs for monitoring the LAI, leaf nitrogen concentration (LNC), and chlorophyll content.

#### 3.4.2. Training Dataset and Test Dataset

To ensure consistency across different viewing zenith angles (VZAs), we employed a uniform dataset partitioning method before modeling [39]. Specifically, the 180 samples were randomly divided into two sets: 70% (n = 126) for training and 30% (n = 54) for testing. This partitioning was maintained across all studies conducted at the same VZA.

#### 3.4.3. Support Vector Machines (SVM)

Machine learning models can be categorized into single models and ensemble models. Among these models, support vector machine (SVM) is particularly effective for the inverse estimation of crop growth parameters [44]. The SVM transforms data into a high-dimensional feature space to establish a linear model and fits a regression function based on this model. SVM can largely overcome issues such as multiple discrete values and overfitting. By choosing an appropriate kernel function, the performance and accuracy of SVM can be enhanced. In this study, we compared the effects of linear, polynomial, and radial basis function (RBF) and Sigmoid kernel functions, concluding that the RBF performed best. The RBF kernel offers advantages such as wide mapping dimensions, fewer parameters to determine, and relatively simple operation [45].

#### 3.4.4. eXtreme Gradient Boost (XGBoost)

Among the ensemble models, boosting and bagging are commonly used. XGBoost, a boosting algorithm, is frequently employed for the inverse estimation of agronomic parameters [46]. XGBoost iteratively combines weak base learners to form stronger learners. To control overfitting, it is crucial to manage model complexity and increase randomness. In this study, the max_depth is chosen as 5, the learning rate is 0.01, and the regular term coefficient is realized by adjusting the alpha and lambda values.

#### 3.4.5. Random Forest (RF)

Random Forest, a popular bagging model in ensemble learning, is extensively used for the inverse estimation of crop growth parameters. This method involves sampling n samples from the original training set using bootstrap resampling, building decision trees for each sample to obtain n modeling results, and finalizing the prediction through voting on all decision tree results. Random Forest is effective for handling large datasets, estimating specific feature variables, managing noise, and providing fast computations [47]. After error analysis and repeated experiments, we selected ntree = 500 and mtry = 3 for model construction.

#### 3.4.6. Evaluating Model Performance

Model performance was evaluated using the using the coefficient of determination (R^2^) and the root mean square error (RMSE). The formulas for R^2^ and RMSE are as follows:(4)$$R^{2}=\frac{\sum_{i=1}^{n}{{(O}_{i}-P_{i})}^{2}}{\sum_{i=1}^{n}{{(O}_{i}-\bar{O_{i}})}^{2}}$$
(5)$$\mathrm{RMSE}=\sqrt{\frac{1}{n}\sum_{i=1}^{n}{{(O}_{i}-P_{i})}^{2}}$$
where P_i_ and O_i_ are the predicted values and observations, respectively, ${\bar{O}}_{i}$ represents the average of the observations, and n is the number of samples.

#### 3.4.7. Flowchart

The workflow chart of this study is illustrated in Figure 8. The main stages include: (1) Data measurement: Ground measurements (LAI) and hyperspectral remote sensing data were collected at specified intervals. (2) Data analysis: The Angle sensitivity of different VI under different experimental conditions after SG smoothing was analyzed. (3) Model and result: The optimal index OPIVI was input into different machine learning models to get the best modeling decision.

## 4. Discussion

### 4.1. The Impact of View Zenith Angle on Band Information and VI

He et al. [48] studied the impact of changes in VZAs on the crop spectral characteristics. They found significant differences in the reflectance values for green and NIR bands at varying VZAs. The reflectance values for green bands are primarily influenced by the absorption of pigments such as chlorophyll a/b, carotenoids, and lutein [49]. Discrepancies in these bands can be attributed to differences in pigment information acquisition at different VZAs. Conversely, NIR band reflectance is mainly influenced by canopy characteristics such as the LAI, biomass, the leaf tilt angle, and canopy distribution patterns [50]. Changes in VZAs influence these canopy characteristics, leading to variations in band emissivity.

In this study, most VIs were constructed at vertical angles, making them sensitive to changes in VZAs [39,43]. Among the nine VZAs, the LAI showed a strong correlation with VIs constructed at near-vertical angles. However, this correlation weakened as the viewing angle increased. This is because larger viewing angles capture canopy characteristics from the upper part of the canopy, while rapeseed plants typically have more leaf area in the lower parts of their canopies during the budding and flowering stages [51]. Thus, spectral information obtained at near-vertical angles is more closely related to the LAI. The angular effect of VIs can affect the stability of monitoring field crops. A new spectral index, the OPIVI, incorporates red, blue, and red edge reflectance into the structure of the NDVI, making it an effective parameter for monitoring the LAI. The advantages of the OPIVI lie in its angle insensitivity to blue and red reflectance and its ability to assess crop growth conditions using red edge parameters. Numerous studies have demonstrated strong relationships between the bands in the OPIVI formula and the LAI, nitrogen nutrient availability, contributing to the establishment of stable monitoring models within certain VZA ranges [33,52]. Including blue band parameters reduces atmospheric correction effects and enhances the estimation of physiological indicators while mitigating the angle effect [20,38]. Consequently, the OPIVI maintains a stable and close relationship with the LAI within the SPP.

Different remote sensing monitoring directions provide different canopy information. Back-scattered signals primarily come from illuminated leaves or branches, while forward-scattered signals mainly come from shaded leaves or branches [9]. When monitoring plant water use efficiency (WUE) and crop LAI, back-scattered observations exhibit stronger robustness and information content than front-scattered ones [53]. In this study, most VIs demonstrated higher ability to monitor LAI at −15° than at vertical or frontal VZAs. Among the back-scattered viewing angles, −15° exhibited optimal performance. Huang et al. [51] and Ratutiainen et al. [54] also monitored leaf chlorophyll content and the LAI using similar VZAs, as spectral information at −15° viewing angles better reflected the vertical gradients of plant growth information. Furthermore, the relationship between different VIs, composed of various band numbers, and the LAI varies with changes in VZAs. Among these three VIs with high correlation and stability, namely EVI-1, the OPIVI, and the REP, all incorporate blue, red, and red edge parameters. This indicates that the stability advantage of OPIVI is not solely attributed to a single band but rather to the combined use of multiple bands, enhancing the accuracy and stability of the OPIVI.

### 4.2. The Impact of Experimental Factors on the Relationship between VIs and LAI

Different experimental treatments on crop canopy structure directly affects the angle sensitivity of VIs. Consequently, VIs exhibit varying angle sensitivities under varying experimental conditions. For example, He et al. [25] found that models performed better during the reproductive stage than the nutrient stage, likely due to stable canopy structure and reduced nitrogen dilution during reproduction. Additionally, the growth direction of leaves (horizontal or upright) influences canopy structure and vertical gradients [55]. In this study, the relationship between VIs and LAI varied with different experimental conditions. For instance, the relationship between VIs and LAI was stronger during the flowering stage compared to the budding stage, which may be attributed to the significant difference in canopy structure between these two stages. Crop nitrogen dilution effects were present from the bolting to flowering stages, and canopy structures reached a stable state during the flowering stage. Thus, there were significant differences in the relationship between VIs and the LAI during these stages. Among these three VIs, EVI-1, REP, and OPIVI, the red band parameter was crucial. Due to pigment absorption by rapeseed flowers during the two growth stages, there were significant differences in reflectance values for the red band, leading to underpinning the monitoring accuracy of EVI-1 depending on the growth stage. Nitrogen fertilization treatments influenced the population density and LAI of rapeseed plants, while cover treatments mainly affected soil water content changes in rapeseed populations [56,57]. Cover treatment had little impact on LAI changes during the late growth stage of rapeseed plants. As a result, correlations between VIs and the LAI were weak under nitrogen fertilization treatments but were stronger under cover treatments, as evidence by the REP parameter. The OPIVI maintained high stability across different growth stages and experimental treatments due to its balanced composition, allowing it to maintain consistent monitoring accuracy under varying experimental conditions. In conclusion, different growth stages, nitrogen fertilization treatments, and coverage methods significantly affect the relationship between the LAI and VIs. Therefore, it is essential to consider these experimental factors when selecting sensitive indices and constructing monitoring models.

### 4.3. The Suitable Algorithm for OPIVI to Monitor LAI

Different types of machine learning algorithms significantly impact the accuracy of monitoring VIs. He et al. [48] used neural network algorithms to estimate wheat LNC based on different VZAs, achieving a high accuracy with an R^2^ value of 0.82. It is important to note that neural networks are single-model machine learning algorithms. In this study, the SVM algorithm, another single-model approach, performed poorly, with an R^2^ value of 0.71. This may be because single models are prone to dataset fragmentation, which affects the monitoring accuracy [58]. Moreover, the main challenge of the SVM lies in determining the kernel functions and related parameters [59]. Due to limitations in parameter selection, such as kernel functions and penalty factors, its application is somewhat restricted. Ensemble algorithm-based machine learning models generally provide higher accuracy in monitoring crop nutrient parameters. For instance, Yuan et al. [60] developed a model to estimate the LAI of rice by combining single spectral and texture indicators, with the RF model achieving the highest R^2^ value of 0.84. In this study, the LAI monitoring model developed by inputting the OPIVI data from three sensitive VZAs into the RF model achieved the highest accuracy, with an R^2^ value of 0.77. This was the highest accuracy among the three machine learning algorithms tested and was significantly improved compared to the single-band linear regression models. This RF algorithm, as an ensemble learning method, demonstrates strong comprehensive application capabilities and high efficiency in utilizing effective information from multiple-angle VIs [61].

This study contributes to the application of multi-angle remote sensing in agriculture for monitoring crop growth and nutrient parameters. We aim to expand our dataset by including more experimental treatments, planting densities, locations, years, and rapeseed varieties to better analyze the angular effects of VIs under different conditions. To further validate the applicability of VIs, it is crucial to conduct analyses on other crops and utilize multiple datasets. Additionally, applying PROSAIL models to simulate additional VZAs and integrating them with various experimental conditions and algorithms will enable further investigation into the anisotropy of crop canopies. In summary, this study provides a more accurate and stable method for monitoring crop LAI from various perspectives.

## 5. Conclusions

The changes in reflectance and VIs with the change in the VZA highlight the importance of considering angle effects in remote sensing for crop LAI monitoring. Acknowledging the vertical distribution of the leaf area in winter oilseed rape, we developed a new index called the OPIVI that captures the dynamic changes in LAI using blue, red, and red-edge parameters. The OPIVI offers a simple and practical method for monitoring LAI in winter oilseed rape. The main conclusions are as follows:(1)Multi-angle observations reveal that the relationship between the back-scatter direction and LAI is stronger than between the vertical and forward-scatter directions. Among the 16VIs tested, the OPIVI shows the highest potential for monitoring across different VZAs, performing best at an elevation angle of −15°.(2)Different experimental factors, such as growth stage, nitrogen fertilizer application, and coverage method, have varying effects on different VIs and the LAI. Notably, the OPIVI maintains a high correlation and angular stability under various experimental conditions.(3)For monitoring model selection, the combination of the RF model with clustering algorithms and multi-angle OPIVI provide optimal results in estimating winter oilseed rape LAI (R^2^ = 0.77; RMSE = 0.38 cm^2^·cm^−2^). This approach significantly improves accuracy compared to single-angle inversion models.

## Figures and Tables

**Figure 1 plants-13-01901-f001:**
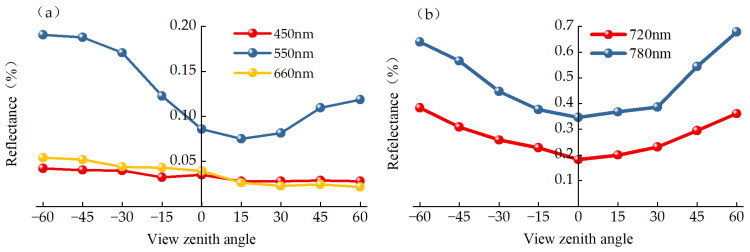
Reflectance changes in blue, green, red, red edge, and NIR bands with different VZAs at the stage of flowering of winter oilseed rape. (**a**) Visible wavelengths. (**b**) NIR wavelengths.

**Figure 2 plants-13-01901-f002:**
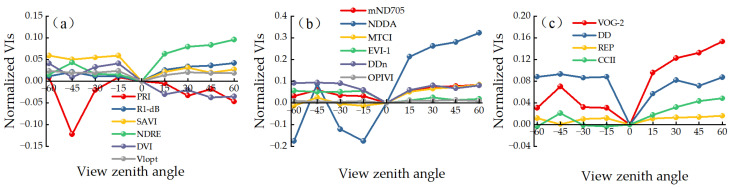
Normalized differences in VI values with respect to nadir between different VZAs within SPP in winter oilseed rape. *X*-axis represents the VZA (negative values represent back-scatter direction, positive values represent forward-scatter direction). *Y*-axis represents the normalized difference. (**a**) Two bands, (**b**) three bands, (**c**) four bands.

**Figure 3 plants-13-01901-f003:**
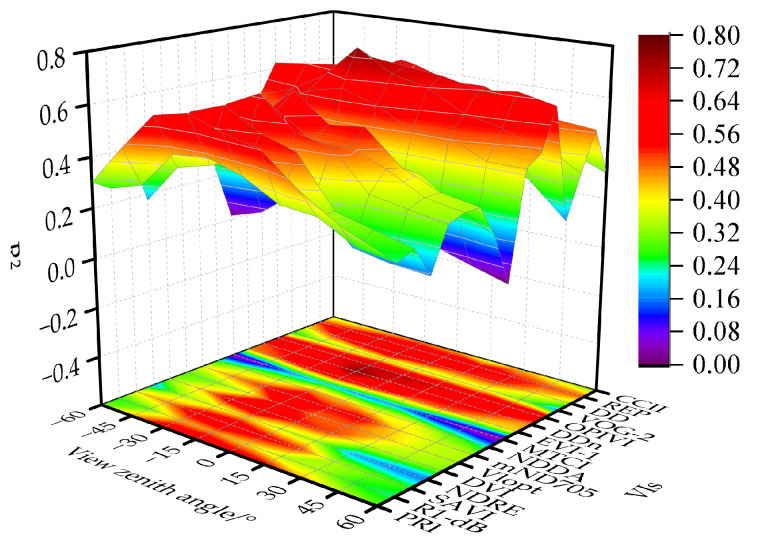
Accuracy of monitoring LAI with different VIs at different VZAs.

**Figure 4 plants-13-01901-f004:**
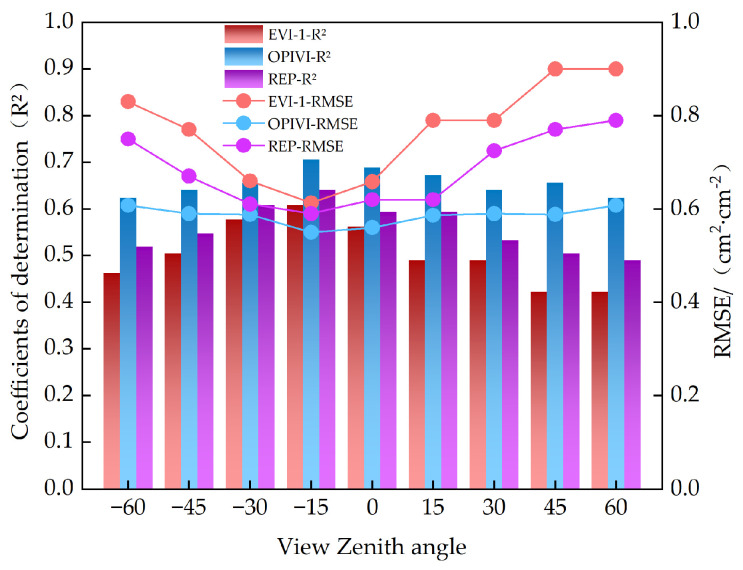
Quantitative relationship between VIs (EVI, OPIVI, and REP) and LAI at different VZAs.

**Figure 5 plants-13-01901-f005:**
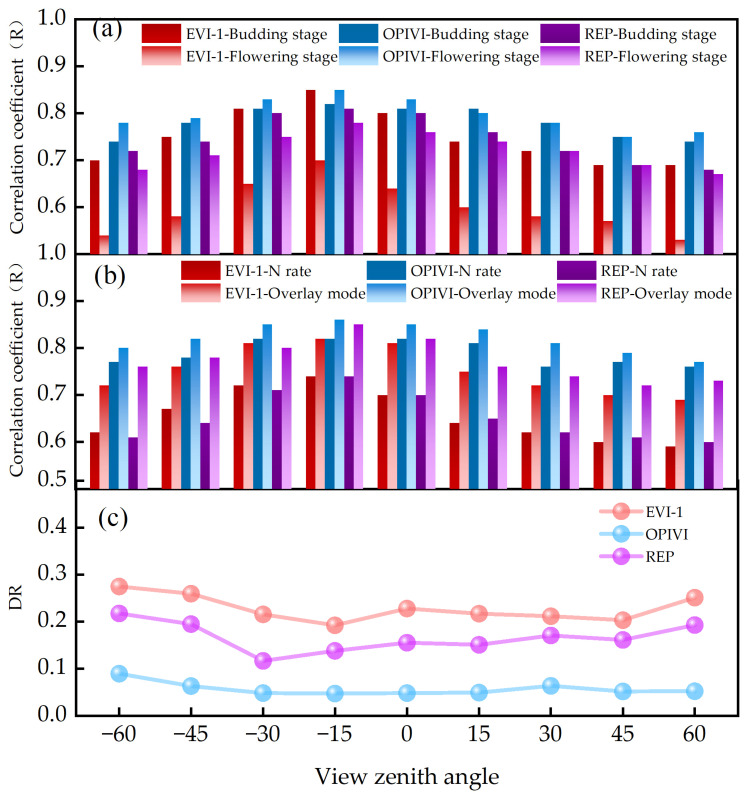
Influence of different experimental factors on the relationship between three VIs and LAI at different VZAs. (**a**) Changes in correlation coefficient of different growth stage VIs at different VZAs. (**b**) Changes in correlation coefficient of different experimental treatment VIs at different VZAs. (**c**) Changes in *DR* value of different VIs at different VZAs.

**Figure 6 plants-13-01901-f006:**
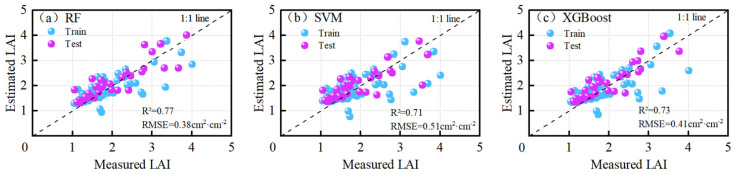
Quantitative relationships of OPIVI to LAI. (**a**) RF, (**b**) SVM, (**c**) XGBoost.

**Figure 7 plants-13-01901-f007:**
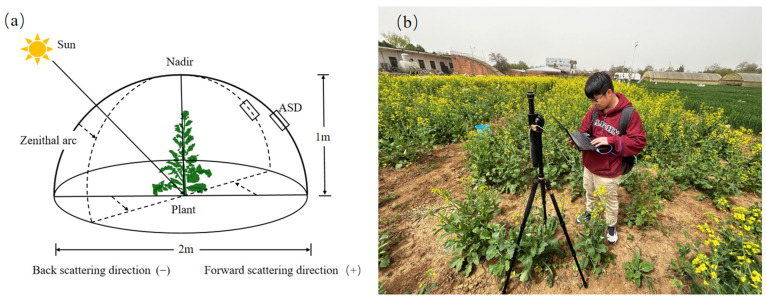
Schematic diagram and instrument for multi-angle hyperspectral measurement. (**a**) Schematic diagram of multi-angle observation at VZA of +60° and +45°. (**b**) Field measurement map.

**Figure 8 plants-13-01901-f008:**
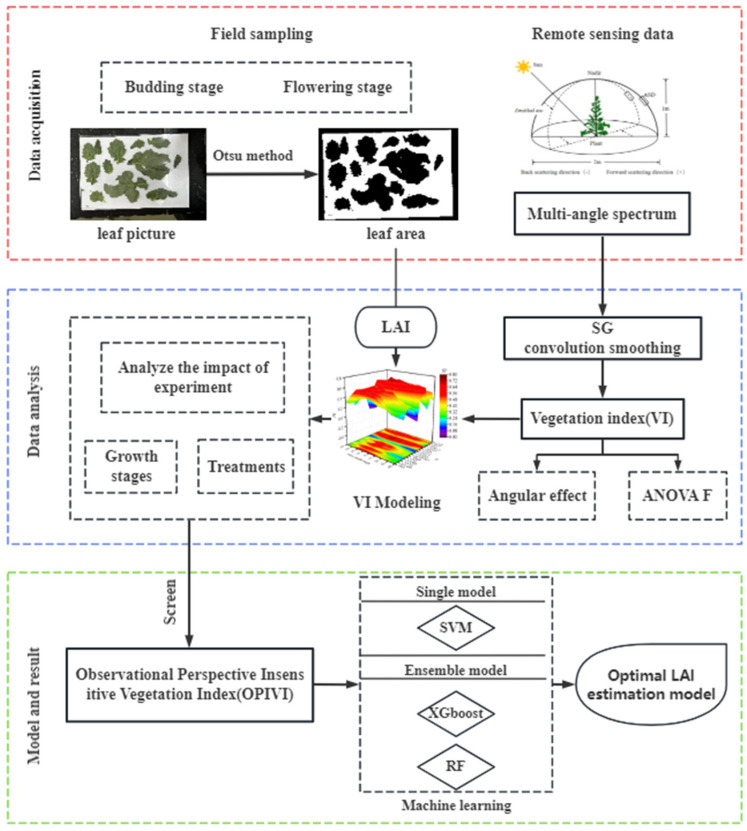
A brief flowchart of this study.

**Table 1 plants-13-01901-t001:** Percentages of change compared to nadir values for extreme VZAs and ANOVA F-ratio values.

Index	−60° vs. Nadir	+60° vs. Nadir	ANOVA F Values (F_8,1620_)	ANOVA F Value (F_2,360_)
	(%)	(%)	(−60°~+60°)	(30°~60°)
**Two bands**				
PRI	1.74	−8.98	3.195 ***	2.805 **
R1-dB	2.33	8.72	7.129 ***	1.509
SAVI	12.52	5.47	3.248 ***	1.160
NDRE	3.23	21.18	5.723 ***	1.298
DVI	8.60	−6.87	2.722 ***	2.180 *
Vlopt	4.79	3.76	2.131 **	0.604
**Three bands**				
mND705	6.64	18.19	6.555 ***	1.226
NDDA	−29.77	95.42	3.901 ***	3.228 **
MTCI	−2.52	18.12	6.119 ***	1.297
EVI-1	11.81	3.90	1.819 *	1.597
DDn	20.22	17.21	1.947 *	1.751
OPIVI	1.60	−6.61	1.735 *	0.444
**Four bands**				
VOG-2	6.35	36.14	5.951 ***	3.447 **
DD	19.31	19.16	4.877 ***	2.062 *
REP	−2.36	3.31	1.877 *	0.760
CCII	−0.70	10.12	3.592 ***	2.499 *

* *p* < 0.05. ** *p* < 0.01. *** *p* < 0.001.

**Table 2 plants-13-01901-t002:** Correlation coefficient (R) between three VIs and LAI at different VZAs. The highest r values for OPIVI are highlighted in bold.

Categories	Sub Datasets	VIs	−60°	−45°	−30°	−15°	0°	15°	30°	45°	60°
Growth stages	Bolting stage	EVI-1	0.70	0.75	0.81	0.85	0.80	0.74	0.72	0.69	0.69
		OPIVI	0.74	0.78	0.81	**0.82**	0.81	0.81	0.78	0.75	0.74
		REP	0.72	0.74	0.80	0.81	0.80	0.76	0.72	0.69	0.68
	Flowering stage	EVI-1	0.54	0.58	0.65	0.70	0.64	0.60	0.58	0.57	0.53
		OPIVI	0.78	0.79	0.83	**0.85**	0.83	0.80	0.78	0.75	0.76
		REP	0.68	0.71	0.75	0.78	0.76	0.74	0.72	0.69	0.67
Treatments	N rates	EVI-1	0.62	0.67	0.72	0.74	0.70	0.64	0.62	0.60	0.59
		OPIVI	0.77	0.78	**0.82**	**0.82**	**0.82**	0.81	0.76	0.77	0.76
		REP	0.61	0.64	0.71	0.74	0.70	0.65	0.62	0.61	0.60
	Overlay mode	EVI-1	0.72	0.76	0.81	0.82	0.81	0.75	0.72	0.70	0.69
		OPIVI	0.80	0.82	0.85	**0.86**	0.85	0.84	0.81	0.79	0.77
		REP	0.76	0.78	0.80	0.85	0.82	0.76	0.74	0.72	0.73

**Table 3 plants-13-01901-t003:** The growth stage of the experiment, solar zenith angle, and azimuth angle.

Year	Date	Stage	Time	Solar Zenith Angle (°)	Solar Azimuth Angle (°)
2023	14 March	Budding	12:30–13:20	48.90–52.77	148.68–167.55
	18 March	Budding	12:30–13:20	50.23–54.23	147.68–167.10
	22 April	Flowering	12:30–13:20	61.52–67.06	135.46–160.89
	25 April	Flowering	12:20–13:10	62.32–68.01	134.18–160.15

**Table 4 plants-13-01901-t004:** The selected vegetation indices used in this study.

Index	Formula	References
Two bands		
PRI (photochemical reflectance index)	(R570 − R531)/(R570 + R531)	[32]
RI-dB (redness index–decibels)	R735/R720	[33]
SAVI (soil-adjusted vegetation index)	1.5 × (R870 − R680)/(R870 + R680 + 0.5)	[34]
NDRE (normalized difference red edge)	(R790 − R720)/(R790 + R720)	[35]
DVI (difference vegetation index)	R860 − R560	[36]
Vlopt (variable light optical properties)	(1 + 0.45) × (R800^2^ + 1)/(R670 + 0.45)	[37]
Three bands		
mND705 (modified normalized difference at 705 nm)	(R750 − R705)/(R750 + R705 − 2 × R445)	[38]
NDDA (normalized difference drought index)	(R680 + R756 – 2 × R718)/(R756 − R680)	[39]
MTCI (meris terrestrial chlorophyll index)	(R754 − R709)/(R709 − R681)	[40]
EVI-1 (enhanced vegetation index-1)	2.5 × (R860 − R645)/(1 + R860 + 6 × R645 − 7.5 × R470)	[20]
DDn (derivative difference normalized)	2.5 × R710 − R660 − R760	[41]
OPIVI (observation perspective insensitivity vegetation index)	(R720 − R450)/(R660 − R450)	This paper
Four bands		
VOG-2 (vogelmann red edge index 2)	(R734 − R747)/(R715 − R726)	[24]
DD (difference vegetation index)	(R749 − R720) − (R701 − R672)	[41]
REP (red edge position)	R700 + 40 × [(R670 + R780)/2 − R700]/(R740 − R700)	[42]
CCII (canopy chlorophyll index integrated)	TCARI/OSAVI	[43]

## Data Availability

The data that support the findings of this study are available from the corresponding author upon reasonable request.

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
