# Peer review of "A New Spectral Index for Monitoring Leaf Area Index of Winter Oilseed Rape (*Brassica napus* L.) under Different Coverage Methods and Nitrogen Treatments"

_plants, 2024, doi:10.3390/plants13141901_

Round 1
Reviewer 1 Report
Comments and Suggestions for Authors
Dear author(s),
I have read your mn (plants-3034753). The following points should be corrected and explained to readers:
Title
1. The title of the mn could be concise. An alternative title can be considered “A new spectral index for monitoring leaf area index of winter oilseed rape under different coverage methods and nitrogen treatments”
Abstract
2. In the 1st sentence (S1), A starter sentence about the importance of the study should be added in the abstract.
3. In the 4th sentence, (S4) please write the scientific name of oilseed rape.
4. S13, the abbreviation “OPIVI” could be written and then its abbreviation can be used.
5. S13, change R to r (correlation)
Keywords
6. the scientific name of oilseed rape could be written,
Introduction
7. In the first parag., change “… but are time-consum” to but “they are time-consum”
8. In the second parag., remove dote before “… making ….”
9. In the second parag., change “Ha-segawa et al. (2010)” to Ha-segawa et al. [8]
10. In the second parag., correct as Stagakis et al. [9]
11. In the third parag., correct as Galvão et al. [22]
12. In the third parag., correct as Verrelst et al. [23]
13. In the third parag., correct as Hovi et al. [24]
14. Please check others and correct them.
15. In the fourth parag., change “China” to “the world and China as well”
M&M
16. In the first parag., experimental design and no. of repetition should be written
17. In the first parag., nitrogen application should be given to readers.
18. In the second parag., change Field-Spec to field-spec
19. In the second parag., change Field Gonimeter System to field goniometer system.
20. In Table 1, date could be written as day/month.
21. In the fourth parag., are three plants enough for the plot area of 24 m2?
22. In Table 2, check the formulas. The last formula could be italicized.
23. Please check eXtreme?
Results
24. In Figure 3, check colors.
25. In Table 3, p (probability) should be italicized. Please check the whole text.
26. In Figure 4, the abbreviations should be explained to readers in the figure caption.
27. In Figure 5, the abbreviations should be explained to readers.
28. In Figure 6, the abbreviations should be explained to readers.
Discussion
29. Correct as He et al. [49]
30. Check the whole text and correct others.
Author Response
We are very grateful for your professional review of our articles, and we are very honored to have the reviewer's recognition of our articles. To facilitate this discussion, we first retype your comments in italic font and then present our responses to the comments.
Comments 1: The title of the mn could be concise. An alternative title can be considered “A new spectral index for monitoring leaf area index of winter oilseed rape under different coverage methods and nitrogen treatments”
Response 1: Thank you for your suggestion! We agree with this comment and have therefore adopted your suggested article title in the revised manuscript.
Comments 2: In the 1st sentence (S1), A starter sentence about the importance of the study should be added in the abstract.
Response 2: Thank you very much for pointing out this issue! We have added a sentence about the importance of this study in the revised manuscript. The specific addition is “Leaf area index (LAI) is a crucial physiological indicator of crop growth. This paper introduces a new spectral index to overcome angle effects in estimating the LAI of crops”.
Comments 3: In the 4th sentence, (S4) please write the scientific name of oilseed rape.
Response 3: Thank you very much for your suggestions! In the revised manuscript we have added the scientific name of oilseed rape: winter oilseed rape (Brassica napus L.).
Comments 4: S13, the abbreviation “OPIVI” could be written and then its abbreviation can be used.
Response 4: Thank you very much for your suggestion! We have added the full name of OPIVI in the revised manuscript: Observation perspective insensitivity vegetation index (OPIVI).
Comments 5: S13, change R to r (correlation)
Response 5: Thank you for pointing this out, and in the revised manuscript we have changed R to r.
Comments 6: The scientific name of oilseed rape could be written.
Response 6: Thank you very much for your suggestions! In the revised manuscript we have added the scientific name of oilseed rape: oilseed rape (Brassica napus L.).
Comments 7: In the first parag., change “… but are time-consum” to but “they are time-consum”.
Response 7: Thank you for your suggestions! We have made changes in the revised manuscript based on your suggestions.
Comments 8: In the second parag., remove dote before “… making ….”
Response 8: Thank you very much for pointing this out! In the revised manuscript we removed the dote before "...making ...." as you suggested ... making ".
Comments 9: In the second parag., change “Ha-segawa et al. (2010)” to Ha-segawa et al. [8].
Response 9: Thank you for your suggestion, corresponding changes have been made.
Comments 10: In the second parag., correct as Stagakis et al. [9]
Response 10: Thank you for your suggestion, corresponding changes have been made.
Comments 11: In the third parag., correct as Galvão et al. [22]
Response 11: Thank you for your suggestion, corresponding changes have been made.
Comments 12: In the third parag., correct as Verrelst et al. [23]
Response 12: Thank you for your suggestion, corresponding changes have been made.
Comments 13: In the third parag., correct as Hovi et al. [24]
Response 13: Thank you for your suggestion, corresponding changes have been made.
Comments 14: Please check others and correct them.
Response 14: We apologize for these problems in the previous manuscript. In the revised manuscript, we have carefully checked all the relevant issues and completed all the corrections according to your suggestions.
Comments 15: In the fourth parag., change “China” to “the world and China as well”
Response 15: Thank you for your suggestion! We very much recognize your suggestion and have rewritten the sentence in the revised manuscript based on your suggestion.
Comments 16: In the first parag., experimental design and no. of repetition should be written
Response 16: Thank you very much for pointing this out! Based on your suggestion, we have added this section to the revised manuscript. The revised content can be found on page 3, lines 111-116 of the revised manuscript
Comments 17: In the first parag., nitrogen application should be given to readers.
Response 17: Thank you very much for pointing this out! Based on your suggestion, we have added this section to the revised manuscript. The revised content can be found on page 3, lines 111-116 of the revised manuscript.
Comments 18: In the second parag., change Field-Spec to field-spec.
Response 18: Thank you very much for pointing out this problem! Based on your suggestion, we have changed Field-Spec to field-spec in the revised manuscript.
Comments 19: In the second parag., change Field Gonimeter System to field goniometer system.
Response 19: Thank you very much for pointing this out! Based on your suggestion, we have changed Field Gonimeter System to field goniometer system in the revised manuscript.
Comments 20: In Table 1, date could be written as day/month.
Response 20: Thank you very much for pointing this out! Following your suggestion, we have changed the time in Table 1 to a day/month format. The changes can be found on page 4, line 150 of the manuscript.
Comments 21: In the fourth parag., are three plants enough for the plot area of 24 m2?
Response 21: Thank you very much for pointing this out. In our study we tried to randomly sample 3-10 plants to represent the level of LAI in each plot, and in the end, based on the results, we found that if we could sample three oilseed rape plants per plot in a triangular shape, the average LAI sampled would be roughly representative of the average level in that plot.
Comments 22: In Table 2, check the formulas. The last formula could be italicized.
Response 22: Thank you very much for pointing this out! Based on your suggestion, we have rechecked and corrected all the formulas in Table 2 and changed the last formula in Table 2 to italicized.
Comments 23: Please check eXtreme?
Response 23: Thank you very much for your suggestion! After our reading of the relevant literature and presentations, we found that eXtreme is correct. the full name of the XGBoost model is eXtreme Gradient Boosting.
Comments 24: In Figure 3, check colors.
Response 24: Thank you very much for your suggestion! After our careful examination, we found no problem with the color of Figure 3 in the manuscript.
Comments 25: In Table 3, p (probability) should be italicized. Please check the whole text.
Response 25: Thank you very much for pointing out this problem! In the revised manuscript we have changed the p to italics in Table 3 and checked the whole table for problems.
Comments 26-28: In Figure 4,5 and 6, the abbreviations should be explained to readers in the figure caption.
Response 26-28: Thank you very much for pointing this out! In the revised manuscript, I have added the full names of the index abbreviations in Table 2. The details can be found on page 5, line 174 of the revised manuscript.
Comments 29: Correct as He et al. [49]
Response 29: Thank you for your suggestion, corresponding changes have been made.
Comments 30: Check the whole text and correct others.
Response 30: Thanks for your careful checks. In the revised manuscript, we carefully checked the full text and revised the questions accordingly.
Reviewer 2 Report
Comments and Suggestions for Authors
It is an interesting approach to conduct field experiments with different coverage methods and nitrogen treatments, including the quantitative analysis of LAI, the testing of VIs under different angles, and the application of machine learning models. The comparative analysis of the three algorithms (SVM, XGBoost, and RF) is effective. By providing specific R² and RMSE values, the reader can easily understand the performance differences, including exact figures for R² and RMSE enhances the credibility and specificity of the analysis.
I recommend publication subject to some revisions, as detailed later in this review.
Here are some comments/queries.
- The statement “monitoring accuracy of the three machine learning algorithms varied significantly” could be made more impactful by quantifying the differences or explaining the range of performance metrics.
- How did you ensure consistency in the division of datasets across different VZAs? Were the same random seeds or stratification methods applied to maintain uniformity?
- The Radial Basis Function (RBF) was selected as the kernel function. Were other kernel functions (e.g., polynomial, sigmoid) considered, and if so, how did they compare in performance?
- What strategies were implemented within XGBoost to prevent overfitting, given its iterative nature? Were any regularization techniques applied?
- How did bootstrap resampling impact the robustness and accuracy of the Random Forest model? Were there any notable variations in model performance with different bootstrap samples?
Minor grammatical adjustments and refinements in phrasing could improve readability. For example, “compare and analyze to obtain the optimal monitoring strategy” could be rephrased as “compared and analyzed to determine the optimal monitoring strategy.”
Phrases like “offers valuable insights for selecting VIs which overcoming angle effect” could be refined to “provides valuable insights for selecting VIs that overcome the angle effect.”
Author Response
General Comments:
It is an interesting approach to conduct field experiments with different coverage methods and nitrogen treatments, including the quantitative analysis of LAI, the testing of VIs under different angles, and the application of machine learning models. The comparative analysis of the three algorithms (SVM, XGBoost, and RF) is effective. By providing specific R² and RMSE values, the reader can easily understand the performance differences, including exact figures for R² and RMSE enhances the credibility and specificity of the analysis.
Response: We are very grateful for your professional review of our articles, and we are very honored to have the reviewer's recognition of our articles. To facilitate this discussion, we first retype your comments in italic font and then present our responses to the comments.
Comments 1: The statement “monitoring accuracy of the three machine learning algorithms varied significantly” could be made more impactful by quantifying the differences or explaining the range of performance metrics.
Response 1: Thank you very much for your suggestion! In Section 3.4, we can see by analyzing Figure 8 that there is a large difference in the accuracy of the three models (RF, SVM, and XGBoost) for monitoring LAI. According to your suggestion, we added a quantitative description of this difference in the subsequent description, including an introduction of R², RMSE of the three models. We hope that our modifications will satisfy you. The specific modifications are shown on page 12, lines 330-336 of the revised manuscript.
Comments 2: How did you ensure consistency in the division of datasets across different VZAs? Were the same random seeds or stratification methods applied to maintain uniformity?
Response 2: Thank you very much for pointing out this issue! In our work, to ensure data consistency between different VZAs, we use the same dataset partitioning method before the modeling process after each VZAs as a way to achieve dataset consistency in the modeling of different VZAs. We hope our explanation is satisfactory to you. Based on your suggestions, we have made responsive additions in the revised manuscript. The specific changes are shown on page 5, lines 175-178 of the revised manuscript.
Comments 3: The Radial Basis Function (RBF) was selected as the kernel function. Were other kernel functions (e.g., polynomial, sigmoid) considered, and if so, how did they compare in performance?
Response 3: Many thanks to the reviewers for their suggestions. In this study, the kernel function of SVM model was chosen as RBF. before that, we compared the performance differences of different kernel functions, including Linear, Polynomial, RBF, Sigmoid and so on. Through our comparison, we found that different kernel functions have more significant effects on the monitoring effectiveness of SVM models, with the lowest R² being only 0.53, and the highest being 0.71 with RBF as the kernel function.In this study, it is more intended to highlight the differences in the accuracies of the three different supervised learning models (SVM, XGBoost, and RF) for monitoring LAI, and thus the accuracies of the different kernel functions of the SVM were not differences in the different kernel functions of SVM were not included in the manuscript. We have added the reason for choosing the RBF kernel function in the revised manuscript, which is described on page 5 lines 185-189 of the revised manuscript. We hope that our response will satisfy you! Thank you again for your useful insights into our research!
Comments 4: What strategies were implemented within XGBoost to prevent overfitting, given its iterative nature? Were any regularization techniques applied?
Response 4: Thank you very much for pointing out this problem. In fact, in our study, we have taken some measures against the overfitting problem of XGBoost. These include controlling the complexity of the model and increasing the randomness. The complexity of the model is realized by controlling the maximum depth of the limiting tree, in this paper max_depth adopts the range of 3-10, and the regular term coefficients are realized by adjusting alpha and lambda; stochasticity is regulated by controlling the learning rate, and a smaller learning rate improves the robustness of the model. In this study, a learning rate of 0.01 was selected after comparative analysis. We hope that our response to you is recognized by you, and the specific modifications can be found on page 5, lines 193-196 of the revised manuscript.
Comments 5: How did bootstrap resampling impact the robustness and accuracy of the Random Forest model? Were there any notable variations in model performance with different bootstrap samples?
Response 5: Thank you very much for pointing this out. First, bootstrap sampling creates multiple subdatasets by repeatedly sampling the original dataset with playback, and each subdataset is used to train a decision tree model. This approach increases the diversity of the models so that each decision tree learns different features and patterns of the data, thus improving the generalization ability of the whole random forest. Also, bootstrap resampling reduces the risk of overfitting. Because each decision tree is independently trained on a different training dataset, the overfitting of the model to the training dataset is reduced. Second, there is a difference in the performance of random forest models constructed using different Bootstrap samples. This is due to the fact that each Bootstrap sample contains slightly different data points and distributions, leading to differences in the structure and predictive performance of the constructed decision trees. However, as the number of models increases, this variation decreases and the overall performance of the models stabilizes. Ultimately, the random forest model integrates the predictions of each decision tree by voting or averaging, resulting in more accurate and robust predictions. We hope that our response to you is recognized by you, and the specific modifications can be found on page 6, lines 198-205 of the revised manuscript.
Comments 6: Minor grammatical adjustments and refinements in phrasing could improve readability. For example, “compare and analyze to obtain the optimal monitoring strategy” could be rephrased as “compared and analyzed to determine the optimal monitoring strategy.”
Response 6: Thanks for your careful checks. In the revised manuscript, we have made improvements to the meaning and grammar of this sentence based on your suggestions. Please refer to line 18 on page 1 for specific changes.
Comments 7: Phrases like “offers valuable insights for selecting VIs which overcoming angle effect” could be refined to “provides valuable insights for selecting VIs that overcome the angle effect.”
Response 7: Thanks for your careful checks. In the revised manuscript, we have made improvements to the meaning and grammar of this sentence based on your suggestions. Please refer to lines 26-27 on page 1 for specific changes.
Reviewer 3 Report
Comments and Suggestions for Authors
I. General comments:
1. Congratulation for the paper, could be very useful for practical application!
2. The research is original, and the objectives are well-defined. Results could provide an advancement of the current knowledge.
3. The paper fit the journal scope proposing a quantitative analysis under nine view zenith angles (VZAs) of the relationship between LAI and multi-angle hyperspectral reflectance from the canopy of winter oilseed rape at various growth stages, nitrogen application levels and coverage methods.
4. The interpretation of the results could be improved completing the presentation of statistic approach results, by carefully identifying the hypothesis and presentation of tests results.
5. The article is written in appropriate way, the data could be completed adequately to the tables.
6. The paper brings a real benefit by publishing after improvement. The work is adequate for advance of current knowledge. The authors do not present a negative result of a valid scientific hypothesis (the approach did not formulate concrete hypothesis, only suggested).
II. Speciffic comments
The manuscript is almost clear, relevant to the topic and well-structured.
The references are relevant, there are not excessive number of self-citations.
The results are almost reproducible based on the details from the paper.
Key words can be synchronized with title and content, even to adjust view zenith angles.
1. Introduction
The aim was „to design field experiments”…, and there are no one picture from experimental plots. Please, optimize.
2. Materials and Methods
2.4.2 Please, refer at few studies (refferences)!
2.4.3 – 2.4.5 I think it is better to explain how were use (setting conditions etc.) machine learning models.
2.4.6 Usually the concept is „coefficient of determination”.
3. Results
3.1 The beginning text can be at method. What is the motivation of „representative feature spectral bands”?
Attention to interpretation of fig.3 and the expression of fig.3, the values of reflectance axis are seriously different!
Table 3 and interpretation of table 3, please complete table 3 with tabulated values for F-test and the hypothesis tested by ANOVA. (Please, cooperate with a specialist in statistic!).
3.2 The title of fig.5 can be improved.
3.4 Attn „of opivi under different”. Please, argue with explanations on statistics.
4. Discussions
OK.
5. Conclusions
OK.
Author Response
We are very grateful for your professional review of our articles, and we are very honored to have the reviewer's recognition of our articles. As you may be concerned, there are several issues in the manuscript that need to be addressed. According to your nice suggestions, we have made extensive corrections to the previous draft, highlighted in red in the revised manuscript. To facilitate this discussion, we first retype your comments in italic font and then present our responses to the comments.
We are very grateful for your professional review of our articles, and we are very honored to have the reviewer's recognition of our articles. As you may be concerned, there are several issues in the manuscript that need to be addressed. According to your nice suggestions, we have made extensive corrections to the previous draft, highlighted in red in the revised manuscript. To facilitate this discussion, we first retype your comments in italic font and then present our responses to the comments.
Comments 1: The aim was „to design field experiments”…, and there are no one picture from experimental plots. Please, optimize.
Response 1: Thank you very much for your suggestions! In the revised manuscript, we have added images of the actual plots in Figure 1. For more details, please refer to page 4, lines 145-149.
Comments 2: 2.4.2 Please, refer at few studies (refferences)!
Response 2: Thank you very much for your suggestion. In order to provide a more scientific basis for our description, we have added references in section 2.4.2. Please refer to page 5, lines 175-178 of the revised manuscript for specific changes
Comments 3: 2.4.3 – 2.4.5 I think it is better to explain how were use (setting conditions etc.) machine learning models.
Response 3: Thank you very much for your suggestions. In order to make our study more informative, we have added more details on computational resources and hyperparameter settings for the machine learning model in the revised manuscript. The revisions can be found in lines 180-205 of the revised manuscript. I hope you are satisfied with our revisions.
Comments 4: 2.4.6 Usually the concept is „coefficient of determination”.
Response 4: Thank you very much for your suggestion! Based on your suggestion, we have changed the full name of R ² in the manuscript to coefficient of determination. The revised content can be found on page 6, line 207 of the revised manuscript.
Comments 5: 3.1 The beginning text can be at method. What is the motivation of „representative feature spectral bands”?
Response 5: Thank you very much for your suggestion, this suggestion will make the content of our articles more convincing. The selection of these five representative bands here is referenced from the study of He et al. (10.1016/j.rse.2015.12.007). Blue (450 ± 16 nm), green (560 ± 16 nm), red (650 ± 16 nm), red edge (730 ± 16 nm), and near-infrared bands (700-2500 nm) are the five representative characteristic spectra in reflectance spectra. Therefore, the five bands of 450 nm, 550 nm, 660 nm, 720 nm, and 780 nm were selected as the characteristic bands in this study. The revised content can be found on page 7, lines 216-223 of the revised manuscript.
Comments 6: 3.1 Attention to interpretation of fig.3 and the expression of fig.3, the values of reflectance axis are seriously different!
Response 6: Thank you very much for your advice. The vertical coordinates of Fig. 3(a,b) represent the reflectivity values of these characteristic bands, and the description of Fig. 3 is done separately for the variations of different bands, which is feasible after our careful verification. We hope that our explanation will satisfy you!
Comments 7: 3.1 Table 3 and interpretation of table 3, please complete table 3 with tabulated values for F-test and the hypothesis tested by ANOVA. (Please, cooperate with a specialist in statistic!).
Response 7: Thank you very much for your suggestion! This suggestion of yours is very helpful to enhance our findings. In the revised manuscript, we have added the explanation for the F-test in Table 3. The revised content can be found on page 8, lines 227-246 of the revised manuscript.
Comments 8: 3.1 The title of fig.5 can be improved.
Response 8: Thank you very much for your advice! According to your suggestion we have optimized the title than Figure 5, the optimized figure title is Accuracy of monitoring LAI with different VIs at different VZAs. We hope that our modification can satisfy you!
Comments 9: Attn “of opivi under different”. Please, argue with explanations on statistics.
Response 9: Thank you very much for your advice! Here we would like to point out the condition of monitoring LAI by inputting OPIVI under the three angles into different models regardless of the experimental design. To avoid ambiguity, we have improved the meaning of this sentence. Please refer to page 12, lines 328-330 of the revised manuscript for specific changes.